# Current Research Status and Implication for Further Study of Real-World Data on East Asian Traditional Medicine for Heart Failure: A Scoping Review

**DOI:** 10.3390/healthcare12010061

**Published:** 2023-12-27

**Authors:** Jeongsu Park, Seongjun Bak, Hongmin Chu, Sukjong Kang, Inae Youn, Hyungsun Jun, Daeun Sim, Jungtae Leem

**Affiliations:** 1Wonkwang University Gwangju Korean Medicine Hospital, Gwangju 61729, Republic of Korea; ig1210@hanmail.net; 2College of Korean Medicine, Wonkwang University, Iksan 54538, Republic of Korea; pss409@naver.com (S.B.); chhn2443@wku.ac.kr (H.C.); 3Wollong Public Health Subcenter, Paju Public Health Center, Paju 10924, Republic of Korea; 4Department of Convergence Technology for Food Industry, Graduate School, Wonkwang University, Iksan 54538, Republic of Korea; sleepnic@naver.com; 5Department of Acupuncture & Moxibustion, National Medical Center, Seoul 06591, Republic of Korea; eknowkey@naver.com; 6Department of Diagnostics, College of Korean Medicine, Wonkwang University, Iksan 54538, Republic of Korea; 7Research Center of Traditional Korean Medicine, College of Korean Medicine, Wonkwang University, Iksan 54538, Republic of Korea

**Keywords:** scoping review, heart failure, East Asian traditional medicine, real-world data, herbal medicine, cohort study

## Abstract

This study used real-world data (RWD) to explore the long-term effects of East Asian traditional medicine (EATM) on heart failure (HF). A comprehensive search was conducted across five databases to identify relevant studies, which were then reviewed using the Arksey and O’Malley scoping review framework. The analysis focused on a descriptive examination of the long-term outcomes associated with EATM intervention. Methodologically, the study explored various aspects, including study subjects, interventions, applied clinical outcomes, and statistical methods. Out of 258 studies, 12 were selected. Eight studies involved patients with HF, while the others used HF as an outcome. Datasets from the National Health Insurance Research Database were used in Taiwan, while electronic medical record data were used in China and Japan. EATM interventions have been found to be associated with lower mortality and readmission rates. One study indicated that an increased dose of Fuzi, a botanical drug, or prompt use of Fuzi after diagnosis led to a decreased mortality hazard ratio. In two studies examining readmission rates, a significant increase was observed in the non-exposed group, with odds ratios of 1.28 and 1.18. Additionally, in patients with breast cancer, the subdistribution hazard ratio for the occurrence of doxorubicin-induced HF was reduced to 0.69. Although cohort studies with survival analysis were common, methodological flaws, such as issues with statistical methods and HF diagnosis, were identified. Despite these challenges, the study observed an association between EATM and improved clinical outcomes in patients with HF, emphasizing the potential of RWD studies to complement randomized controlled trials, especially for longer-term follow-ups. These results provide foundational data for future RWD research.

## 1. Introduction

Heart failure (HF) is a complex clinical syndrome resulting from structural or functional abnormalities of the heart that damage the ability of the ventricles to fill and drain blood [1]. It involves various mechanisms, including myocardial stretching, matrix remodeling, neurohormonal activation, and inflammation [2,3,4]. The prevalence of HF in developed countries is approximately 1–2%, and is increasing owing to population growth and aging [5,6]. In particular, the mortality rate after 5 years of hospitalization for HF, a high-severity disease, is 42.3% [7]. HF arises from factors, such as high blood pressure, dilated cardiomyopathy, coronary artery heart disease, diabetes, and obesity, impairing heart function and leading to complications [1,8,9,10,11]. This multifactorial condition emphasizes the need for diverse approaches based on country-specific variations owing to significant geographical differences in etiology [12]. Although the standard treatment for HF includes drugs, such as angiotensin receptor blockers, beta-blockers, and diuretics, drug therapy has known side effects, including bradycardia and hypotension [1]. In cases of unsuccessful standard treatment, implanted cardioverter defibrillators (ICD), cardiac resynchronization therapy, and heart transplantation are required [1]. However, these interventions are accompanied by potential side effects, such as infection, arteriosclerosis, and thrombosis. Moreover, owing to the scarcity of donors, heart transplantation remains challenging, highlighting the need for complementary treatment strategies to prevent HF progression and manage complications [3,13].

In East Asia, interventions using East Asian traditional medicine (EATM), such as acupuncture, botanical drugs (herbal medicine), pharmacopuncture, and qigong, have been used to treat heart disease [14,15]. Research on heart function in HF has explored the use of acupuncture and moxibustion, with a combination of these interventions and conventional medication showing improvements in left ventricular ejection fraction (LVEF), cardiac output, 6-min walking test scores, and brain natriuretic peptide levels [16]. Botanical drugs have been used in patients with ischemic heart disease and angina pectoris, and Rhodiola is known to be effective in improving electrophysiology and symptoms in patients with ischemic heart disease [17]. Herbal drug injections have been demonstrated to be effective and safe for chronic heart failure (CHF) treatment when used in combination with traditional approaches [18]. However, the clinical characteristics of HF necessitate long-term follow-up, posing challenges for conducting extended clinical trials of EATM [19]. Although many surrogate outcomes have been evaluated in short-term clinical trials, assessing patient mortality, HF-related hospitalization, cardiovascular disease prevalence, and the use of ICD or cardiac resynchronization therapy requires long-term follow-up [19]. Therefore, a different approach is needed to identify the long-term clinical endpoints.

Despite the demonstrated efficacy of EATM interventions in HF, verifying long-term clinical outcomes has been hindered by the lack of resources for extensive clinical trials. Real-world data (RWD), including health insurance claims, hospital electronic medical records (EMRs), and registry data, offer an alternative to exploring the effects of specific interventions on long-term clinical outcomes without conducting clinical trials. In Taiwan, Korea, Japan, and China, RWD studies on the treatment effects of EATM intervention for HF are underway, using various epidemiological research methodologies, such as retrospective cohort or case-control studies based on diverse data sources. Nevertheless, an overview, scoping, and summary of RWD studies on HF have not been conducted. Therefore, we conducted a scoping review to explore RWD studies using EATM interventions to treat HF. This study aimed to explore the long-term effects of EATM intervention. Additionally, we analyzed the characteristics, advantages, and limitations of the RWD study design for HF and suggested future directions for follow-up studies.

## 2. Methods

This study followed the scoping review methodology developed by Arksey and O’Malley [20], chosen for its relevance in the exploration of a new topic involving RWD [21]. The Preferred Reporting Items for Systematic Reviews and Meta-Analyses Extensions for Scoping Reviews guidelines were followed to develop this scoping review protocol [22]. The protocol for the review was recorded by the Open Science Framework and registered on 24 August 2022 (https://osf.io/wrc9d/ (accessed on 24 August 2022)).

### 2.1. Step 1: Checking Study Questions

The research team, comprising clinical research experts, Korean medicine specialists in circulatory diseases, and Korean medicine clinical researchers, conducted a literature search and specified the scope of the subject. The search addressed key inquiries about HF based on RWD. It provided valuable insights by examining the study design, identifying frequently used clinical outcomes, and evaluating the long-term effectiveness and safety of EATM interventions in HF. Additionally, the study scrutinized botanical drug regimens for HF treatment. It aimed to identify specific botanicals, distinguished by their scientific names, used to treat individuals with HF. Overall, these findings enhance our understanding of HF treatment in real-world clinical scenarios.

### 2.2. Step 2: Checking Relevant Studies

This review focused on peer-reviewed studies using RWD-based EATM interventions in patients with HF. The initial literature search was conducted in May 2022, and the final search, using the same strategy, was conducted in September 2023. The following databases were searched: MEDLINE via PubMed, EMBASE via Elsevier, Cochrane Central Register of Controlled Trials, Cumulative Index to Nursing and Allied Health Literature, and Allied and Complementary Medicine. The search strategies were discussed by clinical research experts and Korean medicine doctors, employing a combination of terms, including HF, RWD (“cohort” or “case-control” or “cross-sectional”), and EATM (“Chinese medicine” or “Kampo medicine” or “Korean medicine”). The search strategy incorporated various medical subject heading terms and synonyms, and detailed search formulas are provided in Appendix A.

### 2.3. Step 3: Study Selection

Three authors (JP, SB, and DS) independently performed the study selection. EndNote 20 was used to remove duplicates from the searched publications that included titles and abstracts. For studies identified as potentially relevant, the full text was checked to determine inclusion. All articles were extracted in Excel and categorized as either included or excluded based on predefined criteria, with reviewers providing reasons for the exclusion. Discrepancies were resolved through discussions with other researchers. The inclusion criteria were as follows: (1) longitudinal studies using RWD, such as prospective cohort, retrospective cohort, and case-control studies; (2) studies using EATM intervention; and (3) studies focused on patients with HF or the occurrence of HF. The exclusion criteria were as follows: (1) studies not following a longitudinal design but instead using a snapshot (cross-sectional) research approach, such as assessing medical usage status in a specific year; (2) survey research and biomarker development studies; (3) interventions conducted by non-physicians; and (4) non-RWD research, including randomized controlled trials (RCTs).

### 2.4. Step 4: Charting Data

The pilot data extraction sheet was completed through discussions with the research team. Following several pilot tests, the extracted items included general information about the study, data source and type of RWD study, HF diagnostic criteria, statistical methods for analyzing clinical outcomes, risk factors associated with HF prognosis, co-medications, HF-related clinical presentations, and detailed information about botanical drugs. Data extraction was independently performed by three reviewers (JP, SB, and DS), who cross-checked the data from all studies. Any discrepancies among the reviewers were resolved through discussions with other researchers (JL).

### 2.5. Step 5: Comparing, Summarizing, and Reporting Results

The extracted data were used to establish a comparison, synthesis, and summarization framework. Table 1 presents the characteristics of the included studies and Table 2 presents the research methodologies and statistical analysis techniques. The classification of clinical outcomes is detailed in Table 3, and the effectiveness and safety of the EATM are provided in Table 4. Table 5 and Table 6 present information on the botanical drug used and its components, while specifics regarding usage frequency and regimen analysis are available in the Appendix A.

## 3. Results

### 3.1. Characteristics of the Included Studies

Of the 258 studies identified, 12 were selected (Figure 1), of which 8 were conducted in patients with HF [23,24,25,26,27,28,29,30]. Of the remaining four studies, HF occurrence was the primary outcome in two [31,32] and secondary outcome in the other two studies [33,34]. Among the eight studies involving patients with HF, four included patients with CHF [23,24,26,28], one included patients with acute HF [25], and one included patients with acute decompensated HF [27]. Two studies included unspecified patients with HF [29,30]. Geographically, the studies were distributed as follows: six in China [24,26,27,28,33,34], five in Taiwan [23,29,30,31,32], and one in Japan [25]. The National Health Insurance Research Database was used in Taiwan, whereas in the other two countries, EMR data from hospitals was used (Table 1).

### 3.2. Diagnosis and Patient Selection Criteria, and Statistical Analysis Methods

Regarding methodological and statistical analyses, five studies conducted survival analysis using propensity score matching [23,24,29,31,32]; some included a dose–response analysis to explore causality [23]. Independent *t*-tests or chi-square tests (which did not consider the time-to-event analysis) were used to compare the mean or ratio of clinical indicators [27,28,29,33,34]. Logistic regression analysis was used to identify the factors influencing the occurrence of cardiac events [24,25,33,34]. Regression analysis was also conducted to predict the factors influencing the use of traditional Chinese medicine [27,30,32,33,34]. One study performed multivariate logistic regression analyses and presented nomograms, receiver operating characteristic curves, and clinical decision curve analyses to construct a model for predicting survival [26]. Table 2 presents the covariates used in each study.

### 3.3. Clinical Outcomes

The clinical outcomes used were classified into mortality, cardiovascular events (including HF occurrence), hospitalization, cardiac function, medical cost, safety, and dose–response relation (Table 3).

Although its effect on mortality remains debatable, EATM intervention in patients with HF appears to reduce mortality. In the study by Tai et al., the hazard ratio was 0.99 (95% confidence interval, 0.76–1.27), indicating no significant reduction in mortality [23]. However, other studies reported a lower 5-year mortality rate in the TCM group than in the control group, with a hazard ratio of 0.24 [29] or odds ratio of 0.19 [26]. In patients with myocardial infarction (MI), the odds ratio of cardiogenic death in the control group ranged from 1.34 to 2.64 [33,34]. Regarding hospital visits, the overall TCM use was associated with a reduction in readmission rates. Guan et al. showed that the readmission rate due to HF was significantly lower in a TCM user group [24]. Another study comparing TCM users of Fuzi (a medicinal botanical drug) versus TCM users without Fuzi found no significant differences in admission rates to intensive care units and hospitalization rates due to HF, MI, and stroke [23]. However, the study noted a trend indicating a lower mortality rate associated with the prompt use of Fuzi after the diagnosis of HF and an increase in Fuzi dosage [23]. Among individuals undergoing doxorubicin chemotherapy for breast cancer, TCM use was associated with a lower incidence of CHF [31]. Another study found a lower incidence of HF when botanical drugs were used for over 180 days in patients with hypertension [32]. Among patients with MI, the TCM group had a lower incidence of acute HF [33,34]. Regarding cardiac function, the TCM group showed a greater improvement in NYHA classification and LVEF [24]. In another study, TCM use had no significant effect on LVDD and cardiac index but was associated with improvement in LVEF, CO, every cardiac output, and 6-min walking test scores [28]. Regarding medical costs, expenses for outpatient treatment or hospitalization within one year after HF diagnosis were significantly lower in a TCM user group [29]. No significant adverse events were reported in the available literature (Table 4) [23,27,28].

### 3.4. Botanical Drug (Herbal Medicine) Regimens

Of the 12 studies, 10 reported the use of botanical drugs prescribed in various forms such as decoctions, pills, capsules, and injections for treatment [23,24,27,28,29,30,31,32,33,34]. One study assessed the prevalence of the use of medications that can cause or exacerbate HF [25]. The botanical drugs used in each study are listed in Table 5 and Table 6. Of the 10 studies, *Panax ginseng* C. A. Meyer (Renshen) was used in 8, followed by *Astragalus membranaceus* Bunge (Huangqi) and *Salvia miltiorrhiza* Bunge (Danshen). Additionally, *Glycyrrhiza uralensis* Fisch. (Gancao) was employed in six studies (Table 5 and Appendix A). Detailed information on the prescribed botanical drugs is provided in Appendix A.

## 4. Discussion

In this study, a scoping review method was used to investigate the RWD research status of EATM interventions for HF. Among the 12 selected studies, 8 were conducted in patients with HF [23,24,25,26,27,28,29,30], 2 used HF as the primary outcome [31,32], and 2 used HF as the secondary outcome [33,34]. All interventions were based on traditional botanical drugs, including TCM and Kampo. Mortality was the most frequently used outcome variable [23,24,25,26,27,29,31,33,34]. Overall, the use of EATM was associated with lower mortality and readmission rates. Furthermore, TCM use in patients with other conditions (such as breast cancer, hypertension, and MI) was associated with a lower incidence of HF. Among the 12 studies, 10 confirmed the use of botanical drugs for treatment, with *Panax ginseng* C. A. Meyer being the most used. The incidence of adverse events was not higher in the TCM group.

Regarding cost-effectiveness, one year after the occurrence of HF, the TCM group exhibited lower medical costs and shorter hospital stays. However, five years after the onset of HF, the TCM group experienced higher costs, accompanied by an increased frequency of outpatient visits. Additionally, traditional medicine is commonly used concurrently with standard treatments, making it costlier than groups receiving only standard treatments. Therefore, economic evaluations should consider not only direct medical costs but also direct non-medical costs, quality-adjusted life years, and other relevant factors [35,36]. The criteria for economic evaluation may vary by country, emphasizing the need for additional country-specific research.

Regarding the herbs prescribed for treatment, *Panax ginseng* C. A. Meyer (Renshen), *Astragalus membranaceus* Bunge (Huangqi), and *Salvia miltiorrhiza* Bunge (Danshen) were predominantly used. *Panax ginseng* C. A. Meyer (Renshen), known for its ability to tonify qi and yang, has diverse pharmacological effects, regulates blood glucose and cholesterol levels, and lowers blood pressure, thereby influencing the risk factors associated with cardiovascular diseases [37,38]. Similar to *Panax ginseng* C. A. Meyer, *Astragalus membranaceus* Bunge (Huangqi), which tonifies qi, is reported to have anti-inflammatory, antioxidant, vascular protective, and diuretic effects [39,40,41]. In China, the combination of *Panax ginseng* C. A. Meyer (Renshen) and *Astragalus membranaceus* Bunge (Huangqi) is commonly used for CHF because of the synergistic effects of the combination of the two herbs [40]. *Salvia miltiorrhiza* Bunge (Danshen), known for its blood-activating and stasis-resolving characteristics, is reported to exert cardiac protective effects attributed to its antioxidative, anti-inflammatory, and antiapoptotic properties [42]. Guo et al. conducted data mining to report the prescription rules for herbs commonly used for preserving ejection fraction in HF [43]. All three of these herbs were included, highlighting the primary functions of herbal medicine as qi-replenishing, yang-warming, blood-activating, and diuresis-inducing. This suggests that the appropriateness of considering these three herbs for HF aligns with previous findings and our research findings. Despite the positive therapeutic effects of botanical drugs on cardiovascular health, six studies in the analysis included interventions with concurrent standard treatments. Hence, it is crucial to consider the potential interaction between botanical drugs and standard treatments, rather than attributing effects solely to botanical drugs.

In a previous RCT, the traditional botanical drug Qili Qiangxin capsule reduced N-terminal-pro-brain natriuretic peptide levels, although all-cause mortality was not explored owing to the relatively short duration of the RCT [44]. In contrast, our study employed a retrospective cohort design with a long-term follow-up, allowing for the analysis of all-cause mortality [23,25,26,29,31]. We observed differences in the treatment effects between RWDs and RCTs. Regarding readmission, an RCT by Li et al. on the effect of Qili Qiangxin capsules on HF reported a readmission rate of 3.28% in the treatment group (8 of 244 patients) versus 6.48% in the control group (16 of 247 patients) [44]. However, in an RWD study by Guan et al. who investigated the effect of Shenmai injection on CHF, the readmission rate was 32.37% in the treatment group (146 out of 451 patients) versus 38.93% in the control group (232 of 596 patients) [24]. Although participant baseline characteristics, study settings, and interventions differed slightly, the effect size of TCM treatment varied considerably between RWD and RCT design studies, despite measuring the same outcome variable. Similar differences between RWD and RCT results have been observed in other studies [45,46,47]. RWD studies analyze larger sample sizes over longer periods, catering to a more diverse population, and can yield different effect estimates even when the outcome variable is the same as in existing RCTs. Generalizing and predicting which research design will provide a more significant effect on the intervention between RCT and RWD is challenging. However, unlike RCTs, observational studies cannot establish causality and can only demonstrate an association. Therefore, both research designs should be used complementarily.

Some problems have not been described in detail regarding study subjects, and previous RCTs have been criticized for not providing detailed information on HF diagnoses [48]. A review related to the existing RCTs treating HF has highlighted the ambiguity in distinguishing between HF with reduced (LVEF ≤ 40%) and preserved (LVEF ≥ 50%) ejection fractions [19], which have different treatment strategies and prognoses [49]. Furthermore, the evaluation of cardiac function using the NYHA class or ejection fraction has not been widely used. These indices can significantly influence prognosis and treatment response [49]. Consequently, follow-up studies should explicitly present patient characteristics by providing a detailed diagnosis of individuals with HF and an assessment of cardiac function.

The selection of the exposed and non-exposed groups lacked the appropriate utilization of physical, radiological, and biochemical indices, leading to limitations in assessing comparability between the groups. This limitation might be inherent in claims data, and there could be systematic differences between users and non-users of EATM. Although statistical adjustments were implemented in the analysis stage, such as regression, efforts to minimize bias in subject group selection are crucial. Another issue pertains to the statistical analysis methods. To analyze time-to-event data, survival analysis, such as Cox proportional hazard regression, should be applied. However, several studies used mean comparison (t-test) or frequency comparison (chi-square test) instead, representing a methodological error [28,29,33,34]. Statistical analysis methods suitable for the long-term follow-up of RWD research should be considered. Additionally, study design considerations, such as setting the first diagnosis time as the index date rather than the date of the first administration of the treatment group, should be adopted to avoid immortal bias [50,51]. In addition, as in the study by Tai et al. included in this review, dose–response analysis should be actively applied to explore causality [23].

To overcome the limitations observed in existing observational studies on HF, the following recommendations are proposed. First, regarding health insurance data, the inadequacy of baseline characteristic information that can be used as covariates is noted, while EMR data from hospitals lacks long-term clinical outcome data. Therefore, there is a need to integrate these two data sources in future studies. Additionally, synchronizing pattern identification should be considered as it influences the treatment strategy in RWD studies [52]. Second, conducting dose–response analyses is essential to strengthen the evidence of association. Tai et al.’s study employed the restricted cubic spline method in evaluating dose or treatment initiation time [23]. Whereas previous research categorized participants into quartiles, adopting an analysis method that maximizes the advantages of continuous variable data is crucial. Third, it is emphasized that existing HF clinical trials predominantly used surrogate outcomes such as echocardiography, symptoms, and blood tests. However, outcomes such as cardiogenic death or HF readmission, which could provide insights into long-term prognosis, should be actively incorporated. Lastly, quantitative synthesis faces challenges due to the different outcomes reported in each study. A study on the core outcome set for a clinical trial of TCM for HF is currently underway [53]. However, extending this set is crucial when using RWD. Accumulating evidence with common outcomes offers the potential for future quantitative synthesis in observational studies using RWD.

To our knowledge, this study has the advantage of being the first literature review on RWD-based EATM interventions for HF. This study not only explored the effectiveness, but also presented the applied design, regimen, study subjects, and statistical methods. In addition, the outcomes and covariates used are described in detail to provide fundamental data to those who would conduct RWD studies using EATM for HF in the future. However, because of the inherent limitations of the observational study, this research also has fundamental constraints. First, the study’s findings only suggest an association and do not confirm the efficacy of EATM. Whether the effect is from the botanical drug alone or synergistic with standard treatment is unclear; thus, it requires supplementation through experimental studies. Additionally, the included studies lacked precise dosages or administration details, highlighting the need for more accurate descriptions in further research. Second, analyzing safety or compliance was challenging owing to data limitations. To overcome these limitations, future studies should incorporate health insurance data and hospital EMRs to establish diverse measures for evaluating safety and compliance, including laboratory tests and confirmation of medication adherence. Third, the study’s limitations include its applicability beyond East Asia, as it focuses on traditional medicine for HF in East Asian populations. A comprehensive review is needed to determine if the findings apply universally or have geographic specificity. Fourth, the search strategy, encompassing “Chinese medicine”, “Kampo medicine”, and “Korean medicine”, aimed to investigate overall traditional medicine status. However, the extracted studies only broadly assessed the exposure to traditional medicine or included interventions based on herbs such as injections and decoctions, omitting diverse EATM interventions such as acupuncture and moxibustion. This is considered a limitation of RWD studies because, in retrospective studies based on health insurance data or hospital EMRs, the exact treatment procedure cannot be accurately determined. That the treatment was not mentioned in the paper does not imply non-administration [54,55]. Thus, a more comprehensive evaluation of various East Asian traditional medicine interventions is needed through registries, prospective cohorts, and other forward-looking research.

## 5. Conclusions

This study investigated mortality, readmission rates, cardiac function, and medical expenses as the clinical endpoints of EATM treatment for HF. The review results showed that EATM treatment for HF was associated with reduced mortality and readmission rates. Observational studies using RWD can supplement the limitations of existing RCTs by examining clinical endpoints that require long-term follow-ups. However, RWD studies need to measure objective covariates, such as the detailed classification of HF and cardiac function evaluation, to enhance the reliability of findings. Efforts should be made to reduce bias in the selection of study subjects, and time-to-event analytical statistical methods need supplementation. The data from our study can serve as fundamental information for future research endeavors.

## Figures and Tables

**Figure 1 healthcare-12-00061-f001:**
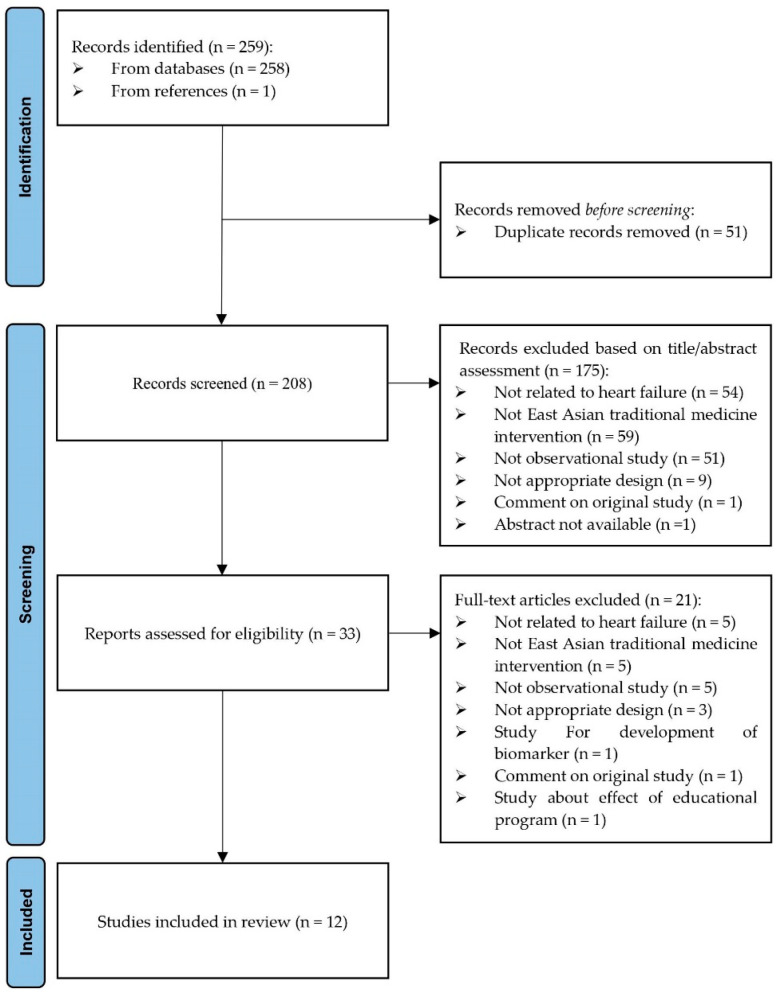
PRISMA flow chart: Selection of studies for scoping review.

**Table 1 healthcare-12-00061-t001:** Characteristics of the included studies with key questions and real-world data sources.

First Author, Year	Country, Period and Follow-Up Duration	Sample Size (Men)	Age (y), Mean ± SD or Median (Range)	Diagnosis, HF Duration or Severity	Intervention	Primary Research Question	Data Source and Research Design
		Ex vs. N-Ex					
Patients with HF							
Tai 2022 [23]	Taiwan (2000–2010) f/u 6 y	921 (272) vs. 921 (272)	68.4 ± 13.4 vs. 67.9 ± 13.8	CHF; ND	TCM (with Fuzi) vs. TCM (without Fuzi)	Effectiveness of Fuzi in combination with routine HF treatment	Population-based propensity score (PS)-matched retrospective nationwide cohort study using NHIRD in Taiwan
Guan 2022 [24]	China (2012–2017) f/u until 2019	451 (204) vs. 596 (332)	67.2 ± 7.0 vs. 67.6 ± 6.7	CHF; (1) Clinical course (months) (Ex) 79.83 ± 53.03 (N-Ex) 64.76 ± 47.98 (2) NYHA classification (II) 71 (15.74): 154 (25.84) (III) 299 (66.30): 339 (56.88) (IV) 81 (17.96): 103 (17.28)	Shenmai injection (more than 7 days) + WM + TCM vs. WM + TCM	Effect of Shenmai injection on the long-term prognosis of patients with CHF	Retrospective cohort study using information systems from seven hospitals in Shandong Province
Komagamine 2021 [25]	Japan (2014–2019)	Non-CEHF-Kampo 437 (179) vs. CEHF-Kampo 30 (20)	(Non-CEHF-Kampo) 80.7 ± 12.3 (CEHF-Kampo) 86.0 ± 7.8	AHF (either new onset or decompensated HF); CCI (Non-CEHF-Kampo) 2.4 ± 1.7 (CEHF-Kampo) 2.0 ± 1.6	CEHF-Kampo vs. Non-CEHF-Kampo	Prevalence of the use of CEHF-Kampo	Retrospective, cross-sectional study using electronic medical records of one hospital
Guan 2021 [26]	China (2012–2017) f/u until 2019	Ex) 228 vs. N-Ex) 196 (men 270; women 142); 308 training set vs. 116 verification set	70 (IQR, 60–75)	CHF; Training set: >36 months 203/308 (65.9%) Verification set: >36 months 60/116 (51.7%)	TCM injection during hospitalization lasting no less than 10 days; Chinese patent medicine or Chinese medicine decoction lasting no less than three months during follow-up	Develop a 5-year survival prediction model for patients with CHF induced by CHD and provide evidence for TCM intervention	Development of a survival prediction model using information systems from seven hospitals in Shandong Province (the same as Guan 2022 [22])
Yu 2019 [27]	China (2015)	TCM) 7400 (3717) non-TCM) 2509 (1347)	TCM) 73 (65–80) non-TCM) 73 (65–80)	Acute decompensated HF; NYHA 2–797 (1.8%): 290 (11.6%) 3–3081 (41.6%): 899 (35.8%) 4–2210 (29.9%): 747 (29.8%).	TCM (oral or intravenous herbal medicine) vs. non-TCM	Practice patterns of TCM, characteristics associated with TCM use, mortality, and bleeding events associated with TCM	Retrospective cohort study using a random sample of 10,004 patients from centralized medical records of 189 Western medical hospitals (China PEACE-Retrospective HF study)
Sui 2018 [28]	China (2015–2017) 6 months f/u	(Ex) 75 (42) vs. (N-Ex) 60 (35)	(Ex) 56.4 ± 9.7 (N-Ex) 58.1 ± 10.2	CHF; NYHA II 40 (53.3): 33 (55.0) NYHA III 23 (30.7): 22 (36.7) NYHA IV 12 (16.0): 8 (13.3) LVEF 24.8 (4.2): 24.5 (4.1)	HM (SQLXF) + WM (including trimetazidine hydrochloride, aspirin, and atorvastatin) vs. WM only	To evaluate the effectiveness and safety of SQLXF	Retrospective cohort study using the hospital information system (one hospital)
Tsai 2017 [29]	Taiwan (2000–2010) 5 y f/u	312 (145): 312 (145)	(Ex) 68.5 ± 12.0 (Non-Ex) 68.3 ± 11.9	ND	TCM (having at least one medical record in a TCM-outpatient clinic and the HF diagnostic code were defined as TCM users) exposed group vs. TCM non-exposed group	Compare medical expenditure and survival with the integration of TCM treatment in patients with HF	Retrospective nationwide cohort study using NHIRD in Taiwan
Tsai 2017 [30]	Taiwan (2000–2010)	TCM user 19,988 (9335) vs. TCM non-user 5476 (3321)	(TCM user) 67.9 ± 13.8 TCM non-user) 72.9 ± 13.5	ND	TCM users (at least one record of TCM treatment 2000–2010) vs. TCM non-users	To explore the frequency and prescription patterns of TCM for patients with HF by analyzing the NHI database from 2000 to 2010 in Taiwan.	Retrospective population-based study using NHIRD in Taiwan
HF as a primary outcome							
Huang 2019 [31]	Taiwan (1997–2010) (f/u until 2013 at least 3 y)	Only women (A) 24,457 (B) 24,457	(A) 51.9 ± 11.2 (B) 52.9 ± 11.1	NA (breast cancer)	TCM exposed (30 days within three months after the diagnosis of breast cancer) + WM (doxorubicin, Herceptin, tamoxifen) vs. non-TCM + WM	Effect of TCM in reducing the risk of HF in patients with breast cancer receiving doxorubicin treatment	Retrospective nationwide cohort study using NHIRD in Taiwan
Liu 2022 [32]	Taiwan (2000–2017)	(A) 8912 (4765) (B) 8912 (4721)	(A) 50.3 ± 11.6 (B) 50.5 ± 11.9	NA (HTN)	TCM users (at least 30 days after the diagnosis of HTN) + WM vs. non-TCM + WM	Effect of TCM in reducing the risk of HF in patients with hypertension	Population-based propensity score-matched retrospective nationwide cohort study using NHIRD in Taiwan
HF as a secondary outcome							
Guohua 2018 [33]	China (2014–2015 June) f/u until 2015 September	479 (259) (TCM injection) High-Ex 168; Low-Ex 178; Non-Ex 133 (TCM patent and decoction) High-Ex 101; Medium Ex 72; Low-Ex 76; Non-Ex 96	67 ± 10	(Patients with AMI + DM) Killip class 1–92 (19.21%) 2–200 (41.75%) 3–129 (26.93%) 4–58 (12.11%)	(Admission stage) TCM injection (main treatment: high Ex group > 14 d; low Ex group 7–13 d) + WM vs. WM (follow-up stage) HM (patent medicine or decoctions: low Ex 28 d–3 m; medium Ex: 3–6 m; high Ex > 6 m) + WM vs. WM	Investigate the effect of TCM on the clinical endpoint of patients with AMI + DM in real-world practice	Retrospective cohort study using information systems from seven hospitals in Shandong Province
Wulin 2018 [34]	China (2014–2015 June) f/u until 2015 September	1596 (977) (TCM injection) High-Ex 561; Low-Ex 622; Non-Ex 413 (TCM patent and decoction) High-Ex 426; Medium Ex 243; Low-Ex 253; Non-Ex 288	66 ± 11	(patients with AMI) Killip class 1–352 (22.06%) 2–734 (45.99%) 3–382 (23.93%) 4–128 (8.02%)	(Admission stage) TCM injection (main treatment: high Ex group > 14 d; low Ex group 7–13 d) + WM vs. WM (follow-up stage) HM (patent medicine or decoctions: low Ex 28 d–3 m; medium Ex 3 m–6 m; high Ex > 6 m) + WM vs. WM	Investigate the effects of tonifying Qi and activating blood circulation on endpoint events in patients with MI	Retrospective cohort study using information systems from eight hospitals in Shandong Province

AMI, acute myocardial infarction; CCI, Charlson Comorbidity Index; CEHF-Kampo, Kampo medications that can cause or exacerbate heart failure; CHD, coronary heart disease; CHF, chronic heart failure; DM, diabetes mellitus; Ex, exposed group; f/u, follow-up; HF, heart failure; HM, herbal medicine; HTN, hypertension; IQR, interquartile range; N-Ex, non-exposed group; NA, not applicable; ND, not described; NHIRD, National Health Insurance Research Database; NYHA, New York Heart Association; QOL, quality of life; SQLXF, Shen Qi Li Xin formula; TCM, traditional Chinese medicine; WM, Western medicine; LVEF, left ventricular ejection fraction.

**Table 2 healthcare-12-00061-t002:** Diagnosis and patient selection criteria, and statistical analysis methods with considered covariates.

First Author, Year	Diagnostic Criteria for HF	Patient Selection	Statistical Method	Covariates		
				Risk Factors and Comorbid Disease for HF	Medication and Treatment	Cardiac Function and Symptom, Imaging, Laboratory
Patients with HF						
Tai 2022 [23]	ICD-9-CM	Patients with chronic HF were identified between 1 January 2000, and 31 December 2010, according to the ICD-9-CM code; 776 patients diagnosed with cancer before HF were excluded.	(1) Survival analysis after PSM using proportional subdistribution hazard regression (2) Dose–response analysis using the RCS function	AF, AFL, CVD, chronic liver disease, CKD, COPD, DM, HTN, hyperlipidemia, IHD	ACEi/ARBs, amiodarone, aspirin, BB, CCBs, clopidogrel, colchicine, digoxin, diuretics, statin, uricosuric agents, XOI, warfarin, other lipid-lowering agents	ND
Guan 2022 [24]	“Chinese HF Diagnosis and Treatment Guidelines” (2014 CHF diagnosis) issued by the Chinese Medical Association Cardiovascular Branch	(Inclusion) (1) consistent with the diagnostic criteria; (2) age range 45–75 years and (3) NYHA class Ⅱ, Ⅲ, or Ⅳ. (Exclusion) (1) AHF; (2) having received CABH or cardiac resynchronization therapy; (3) accompanied by non-CVD, such as malignant tumors, mental illnesses, and severe liver and kidney dysfunction	(1) Survival analysis using Cox regression without PSM (HR not presented) (2) Multivariate logistic regression analysis on cardiac death and HF readmission	The course of CHF, residence; (comorbid disease) A-fib, cardiomyopathy, CHD, DM, HTN,	ACEi/ARBs, ARA, BB, diuretics	(Cardiac function) NYHA
Komagamine 2021 [25]	Causality of CEHF-Kampo was assessed using Naranjo criteria	Aged ≥18 years who were hospitalized for acute HF	Univariate/multivariate logistic regression on predictive factors associated with HF	CCI, drug adherence, drug-induced, nursing home resident; (comorbid disease) anemia, arrhythmias, cardiac ischemia, CKD, COPD, dementia, DM, diet, HF, HTN, IHD, infection, stroke	ACEi/ARBs, BB, loop diuretics, NSAIDs, spironolactone, digoxin	(Cardiac function) LVEF (Laboratory) BNP, Hb, Cr, BUN, Na, K
Guan 2021 [26]	“Chinese HF Diagnosis” and Treatment Guidelines” 2014 CHF diagnosis	(Inclusion criteria) (1) consistent with the diagnostic criteria; (2) age range 45–75 years; (3 NYHA class II, III, or IV; (4) CHF induced by CHD; (5) EF < 50%; (6) follow-up time ≥ 5 years. (Exclusion criteria) (1) AHF; (2) patients who underwent CABG or cardiac resynchronization therapy; (3) associated with no cardiovascular events, such as malignant tumors and psychosis; (4) severe liver and kidney dysfunction; (5) missing data	(1) Multivariate logistic regression analysis to construct the prediction model and nomogram (Lasso regression method was used to filter the best predictor) (2) ROC curve and clinical decision curve analysis to evaluate the utility of the model	Weight, days in hospital, course of CHF; (comorbid disease) arrhythmia, CVD, digestive system disease, DM, HTN, hyperlipidemia, kidney disease, peripheral vascular disease, respiratory disease, and thyroid disease.	ACEi/ARB, antagonists, anticoagulant drugs, aspirin, BB, Ca, cedilanid, clopidogrel, digoxin, diuretics, nitrates, spironolactone, statins, trimetazidine	(Cardiac function) HR, SBP, DBP, NYHA, EF, LVEDD (Laboratory) NT-proBNP, K, Na, Cr, ALT, T-BIL, TG, HDL-C, LDL-C, glucose, APTT, Hb
Yu 2019 [27]	ICD 10 (I50.xx, I11.0x, I13.0x, or I13.2x)	Patients aged ≥18 years were hospitalized between 1 January 2015, and 31 December 2015, with a discharge diagnosis of HF. (exclusion) patients with a principal admission diagnosis of acute myocardial infarction	(1) Descriptive analysis of TCM usage (frequency, duration, timing, correlation between each TCM usage, patient and hospital level characteristics of TCM usage) (2) Hierarchical logistic regression for a factor associated with TCM usage	ICU admission, insurance, facility/region/location of hospital; (comorbid disease) A-fib, anemia, cancer, CKD, COPD/asthma, CHD, dyslipidemia, DM, HTN, PVD, stroke, valvular disease	ND	(Cardiac function/symptom) cardiogenic shock, chest pain, dyspnea (rest/exertion/nocturnal), orthopnea, jugular vein distension, S3, rale, hepatojugular reflux, edema, HR, SBP, DBP, NYHA, (Imaging) chest X-ray, CT scan, echocardiogram (LVEF) (Laboratory) BNP/NT-proBNP, BUN, Cr, Na, K, T-chol, glucose, Hb,
Sui 2018 [28]	CHF and symptoms of the NYHA functional class (II–IV)	(Exclusion) severe valve stenosis; cardiomyopathy (hypertrophy or restriction); pericarditis (constriction), myocarditis; AMI; cardiogenic shock; various malignancies, trauma, connective tissue disease, pregnancy and breastfeeding; severe liver, and renal insufficiency	Independent *t*-test (continuous variables) and Chi-square test (categorical variables)	Alcohol, BMI, CHF family history, smoking; (comorbid disease) DM, HTN, stroke	ACEi/ARBs, antithrombotic agents, diuretics, digoxin	(Cardiac function) NYHA, BP, HR, LVDD, LVEF, CO, ECO, CI, 6 MWT
Tsai 2017 [29]	ICD-9-CM 428	(Inclusion) the population with HF (n = 29, 552) had at least two ambulatory visits or one inpatient claim with a diagnosis of ICD-9-CM code 428 from 2000 to 2010. (Exclusion) lost to follow-up in the NIH program for >1 year or had received TCM treatment before the initial date of HF	Independent *t*-test (continuous variables) and chi-square test (categorical variables); multivariable Cox-proportional hazard regression analysis (5-year survival rate) with 1:1 frequency matching case cohort	(At least 2 ambulatory claims or 1 inpatient claim) CAD, COPD, DM, hyperlipidemia, stroke	ND	(Cardiac function and symptoms) pulmonary edema and respiratory failure
Tsai 2017 [30]	ICD-9-CM 428	(Inclusion) aged at least 20 years who were newly diagnosed with HF (ICD-9-CM 428) from 1 January 2000 to 31 December 2010	(1) Compared demographic characteristics between TCM users vs. TMC non-users (2) Multivariable logistic regression on prediction factors of TCM usage (odds ratio)	CAD, COPD, DM, HTN, stroke	NA	NA
HF as a primary outcome						
Huang 2019 [31]	NA (Breast cancer)	Women, aged 20–80 years and newly diagnosed with breast cancer from 1997 to 2010 (ICD-9-CM 174.X). (Exclusion) patients diagnosed with cancer and CHF before the index day. Patients without doxorubicin medication.	“Fine and Gray” regression hazard model and gray test after PSM	Income, residence; (comorbid disease) A-fib, CKD, CVD, DM, dyslipidemia, HTN, IHD	ACEI/ARBs, alpha-blocker, BB, CCB, diabetic drugs, direct vasodilator, loop diuretics, potassium-sparing diuretics, statins, thiazide diuretics; (breast cancer medication) Herceptin, tamoxifen	(Cardiac function) LVEF
Liu 2022 [32]	ICD-9-CM 401–405; ICD-10-CM 110–115 (Hypertension) ICD-9-CM 428; ICD-10-CM 150 (HF)	(Inclusion) among patients with HTN who were followed up for >3 months, those who had received TCM for at least 30 days. Patients with HF should have been admitted to the hospital at least once or visited outpatient clinics at least three times within a year. (Exclusion) hyperthyroidism, beriberi, anemia, COPD, depression	(1) Log-rank test with Kaplan–Meier curve (2) Multivariable Cox-proportional hazard regression analysis with covariates after PSM	Income, residence; (comorbid disease) CHD, CVD, DM, hyperlipidemia, PAOD	ACEi/ARB, alpha blockers, BB, CCBs, diuretics	NA
HF as a secondary outcome						
Guohua 2018 [33]	NA (AMI + DM patients)	(Inclusion) (a) with AMI and DM; (b) hospitalized from 1 January 2008 to 31 December 2014. (exclusion) (a) with previous MI (patients with recurrent MI were not excluded); (b) who had received PCI or CABG; (c) without contact information or current address; (d) who had serious non-CVD	Independent *t*-test for continuous data and chi-square test for categorical data; multivariate logistic regression to analyze TCM usage and clinical outcome	Arrhythmia, DM, HTN	ACEI/ARB, anti-MI drugs, anti-thrombosis drugs, lipid-lowering drugs,	(Cardiac function) Killip class, stage of development, ST-segment elevation,
Wulin 2018 [34]	NA (AMI patients)	(Inclusion) patients aged 40–88 years with a diagnosis of AMI based on standard criteria. (Exclusion) MI did not recur (but those with recurrent MI were included); previously received PCI or CABG; contact information is unavailable; comorbid diseases of non-cardiovascular origin, such as malignant tumors or mental illness.	Independent *t*-test for continuous data and chi-square test for categorical data; multivariate logistic regression to analyze TCM usage and clinical outcome	Arrhythmia, DM, dyslipidemia, HTN	ACEis/ARBs, antithrombotic drugs, anti-myocardial ischemia drugs, lipid-lowering drugs	(Cardiac function and symptoms) Killip class, stage of development, ST-segment elevation

6 MWT, 6-min walking test; A-fib, atrial fibrillation; ACEi, angiotensin-converting enzyme inhibitor; AF, atrial fibrillation; AFL, atrial flutter; AHF, acute heart failure; ALT, alanine trans-aminase; AMI, acute myocardial infarction; APTT, activated partial thromboplastin time; ARB, angiotensin II receptor blocker; ARA, aldosterone receptor antagonist; BB, beta-blockers; BMI, body mass index; BP, blood pressure; BUN, blood urea nitrogen; Ca, calcium; CABG, coronary artery bypass grafting; CAD, coronary artery disease; CCBs, calcium channel blockers; CCI, Charlson Comorbidity Index; CEHF-Kampo, Kampo medications that can cause or exacerbate heart failure; CHD, coronary heart disease; CHF, chronic heart failure; CI, cardiac index; CKD, chronic kidney disease; CO, cardiac output; COPD, chronic obstructive pulmonary disease; Cr, creatinine; CT, computed tomography; CVD, cerebrovascular disease; DBP, diastolic blood pressure; DM, diabetes mellitus; ECO, every cardiac output; EF, ejection fraction; Hb, hemoglobin; HDL-C, high-density lipoprotein cholesterol; HR, heart rate; HTN, hypertension; HF, heart failure; ICD, International Classification of Disease; ICD-9-CM, the International Classification of Diseases, Ninth Revision, Clinical Modification; ICD-10-CM, International Classification of Diseases, Tenth Revision, Clinical Modification; ICU, intensive care unit; IHD, ischemic heart disease; LDL-C, low-density lipoprotein cholesterol; LVDD, left ventricular diastolic dysfunction; LVEDD, left ventricular end-diastolic diameter; LVEF, left ventricular ejection fraction; MI, myocardial infarction; NA, not applicable; Na, sodium; ND, not described; NSAID, nonsteroidal anti-inflammatory drug; NT-proBNP, N-terminal pro-brain natriuretic peptide; NYHA, New York Heart Association; PAOD, peripheral arterial occlusive disease; PCI, percutaneous coronary intervention; PSM, propensity score matching; PVD, pulmonary vascular disease; RCS, restricted cubic spline; ROC curve, receiver operating characteristic curve; SBP, systolic blood pressure; T-BIL, total bilirubin; T-chol, total cholesterol; TCM, traditional Chinese medicine; TG, triglyceride; XOI, xanthine oxidase inhibitor.

**Table 3 healthcare-12-00061-t003:** Types of clinical outcomes used in included studies.

Classification		Tai 2022 [23] (Taiwan)	Guan 2022 [24] (China)	Komagamine 2021 [25] (Japan)	Guan 2021 [26] (China)	Yu 2019 [27] (China)	Sui 2018 [28] (China)	Tsai 2017 [29] (Taiwan)	Tsai 2017 [30] (Taiwan)	Huang 2019 [31] (China)	Liu 2022 [32] (Taiwan)	Guohua 2018 [33] (China)	Wulin 2018 [34] (China)
Diagnosis	Patients	CHF	CHF	AHF (either new onset or decompensated HF)	CHF	Acute decompensated HF	CHF	HF (Not specified)	HF (Not specified)	Breast cancer	HTN	AMI + DM	MI
Composite outcome	Composite cardiovascular outcomes	Death, stroke, MI											
	Re-myocardial infarction and Stroke											O	O
Mortality	All-cause mortality	O		O	5 y			5 y		O			
	Cardiogenic death		O									O	O
	In-hospital mortality or treatment withdrawal					O							
	Death from other diseases		O										
Cardiovascular event	Acute coronary syndrome		O										
	Occurrence/cumulative incidence of HF			Kampo induced HF						O	O	O	O
	Post infarction angina											O	O
	Arrhythmia											O	O
	Cardiac shock											O	O
	Revascularization											O	O
	Hypotension											O	
	Cardiac arrest											O	
Hospitalization	ICU hospitalization	O											
	HF-related hospitalization	O											
	MI-related hospitalization	O											
	Stroke-related hospitalization	O											
	HF-related readmission		O										
	Rehospitalization											O	O
Cardiac function (surrogate outcome)	NYHA classification		O										
	6 MWT						O						
	LVEF		O				O						
	LVDD						O						
	Cardiac output						O						
	Every cardiac output						O						
	Cardiac index						O						
	NT-proBNP		O										
Medical cost	Outpatients care visits							O					
	Hospital length of stay (days)							O					
	Outpatient care cost							O					
	Hospitalization cost							O					
Safety	Adverse events						O						
	Cardiac arrhythmia-related hospitalization	O											
	In-hospital bleeding					O							
Dose–response relation		A *											
Others						B *			B *				

The outcome used in each study is denoted by an ‘O’. 6 MWT, 6-min walking test; A *, dosage of Fuzi, and their association with mortality and cardiovascular events; AHF, acute heart failure; AMI, acute myocardial infarction; B *, commonly used TCM and a predictive factor for the utilization of TCM; CHF, chronic heart failure; DM, diabetes mellitus; HF, heart failure; HTN, hypertension; ICU, intensive care unit; LVEF, left ventricular ejection fraction; LVDD, left ventricular diastolic dysfunction; MI, myocardial infarction; NYHA, New York Heart Association; NT-proBNP, N-terminal pro-brain natriuretic peptide.

**Table 4 healthcare-12-00061-t004:** Effectiveness of East Asian traditional medicine intervention in included studies.

Category	Outcome	Intervention	Study ID	Effect (TCM Exposed Group vs. TCM Non-Exposed Group)
Mortality	All-cause mortality	Fuzi + HM	Tai 2022 [23]	98/921 (10.6%) vs. 153/921 (16.6%), HR = 0.99 (95% CI, 0.76–1.27)
		TCM + WM	Huang 2019 [31]	4726/24,457 (17.5%) vs. 7152/24,457 (29.2%), *p* < 0.0001
	Cardiogenic death	SMI + WM + TCM	Guan 2022 [24]	50/451 (10.2%) vs. 92/596 (15.4%), *p* = 0.04
		TCM Inject (admission) or HM (f/u) + WM	Guohua 2018 [33]	(During hospitalization) OR 2.04 in Non-ex (*p* < 0.001) (During follow-up) OR 1.34 in Non-ex (*p* = 0.01)
		TCM Inject (admission) or HM (f/u) + WM	Wulin 2018 [34]	(During hospitalization) OR 2.64 in Non-ex (*p* < 0.001) (During follow-up) OR 1.43 in Non-ex (*p* < 0.001)
	Death from other diseases	SMI + WM + TCM	Guan 2022 [24]	2/451 (0.3%) vs. 33/596 (5.5%), *p* < 0.001
		Kampo medications that may cause or exacerbate HF (CEHF)	Komagamine 2021 [25]	Non-CEHF 40/407 (9.8%) vs. 4/30 (13.3%) non-statistical analysis
	In-hospital mortality or treatment withdrawal	TCM +WM	Yu 2019 [27]	242/7400 (3.3%) vs. 100/2509 (4.0%), OR = 1.05 (95% CI, 0.84–1.39)
	5-year mortality	TCM	Guan 2021 [26]	TCM group Odds ratio 0.19 (0.09–0.37)
		TCM	Tsai 2017 [29]	TCM all group HR 0.24 (0.16–0.35) (compensated group) HR 0.14 (0.07–0.28) (decompensated group) HR 0.32 (0.20–0.52)
Hospital visit	ICU/CCU admission	Fuzi + HM	Tai 2022 [23]	160/921 (17.4%) vs. 249/921 (27.0%), HR = 0.96 (95% CI, 0.84–1.11)
	HF admission	Fuzi + HM	Tai 2022 [23]	187/921 (20.3%) vs. 266/921 (28.9%), HR = 1.01 (95% CI, 0.84–1.22)
	MI admission	Fuzi + HM	Tai 2022 [23]	104/921 (11.3%) vs. 169/921 (18.4%), HR = 0.93 (95% CI, 0.72–1.19)
	Stroke admission	Fuzi + HM	Tai 2022 [23]	129/921 (14.0%) vs. 194/921 (21.1%), HR = 1.01 (95% CI, 0.80–1.26)
	HF re-admission	SMI + WM + TCM	Guan 2022 [24]	146/451 (32.4%) vs. 232/596 (38.9%), *p* = 0.03
	Re- hospitalization	TCM Inject (admission) or HM (f/u) + WM	Guohua 2018 [33]	(During follow-up) OR 1.28 in Non-ex (*p* = 0.03)
		TCM Inject (admission) or HM (f/u) + WM	Wulin 2018 [34]	(During follow-up) OR 1.18 in Non-ex (*p* = 0.003)
HF occurrence	HF by doxorubicin	TCM + WM	Huang 2019 [31]	Adjusted sHR 0.69 (95% Cl, 0.62–0.76), *p* < 0.001 (subgroup) Pugongying vs. non-TCM 0.65 (95% CI, 0.41–1.04) Jianghuang vs. non-TCM 0.56 (95% CI, 0.30–1.04) Baihuasheshecao vs. non-TCM 0.29 (95% Cl, 0.15–0.56) Huangqin vs. non-TCM 0.64 (95% CI, 0.48–0.84)
	HF	TCM + WM	Liu 2022 [32]	Adjusted HR 0.85 (95% CI, 0.85 0.74–0.98), *p* = 0.03 (Subgroup) TCM use 30–180 days vs. non-TCM 1.02 (95% CI, 0.87–1.19), *p* = 0.84 TCM use more than 180 days vs. non-TCM 0.65 (95% CI, 0.53–0.79), *p* < 0.001
	Acute HF	TCM Inject (admission) or HM (f/u) + WM	Guohua 2018 [33]	(During hospitalization) OR 1.08 in Non-ex (*p* = 0.67) (During follow-up) OR 1.41 in Non-ex (*p* = 0.03)
		TCM Inject (admission) or HM (f/u) + WM	Wulin 2018 [34]	(During hospitalization) OR 1.04 in Non-ex (*p* = 0.71) (During follow-up) OR 1.34 in Non-ex (*p* = 0.002)
Cardiovascular event prevention	Composite CV outcomes	Fuzi + HM	Tai 2022 [23]	339/921 (36.8%) vs. 466/921 (50.6%), HR = 0.96 (95% CI, 0.84–1.11)
	Composite CV outcomes (Re-infarction and stroke)	TCM Inject (admission) or HM (f/u) + WM	Guohua 2018 [33]	(During hospitalization) OR 1.07 in Non-ex (*p* = 0.88) (During follow-up) OR 1.84 in Non-ex (*p* < 0.001)
		TCM Inject (admission) or HM (f/u) + WM	Wulin 2018 [34]	(During hospitalization) OR 0.82 in Non-ex (*p* = 0.40) (During follow-up) OR 1.35 in Non-ex (*p* < 0.001)
	ACS	SMI + WM + TCM	Guan 2022 [24]	6/451 (1.9%) vs. 8/596 (1.3%), *p* = 0.95
	Severe arrhythmia	TCM Inject (admission) or HM (f/u) + WM	Guohua 2018 [33]	(During hospitalization) OR 1.35 in Non-ex (*p* = 0.20) (During follow-up) OR 1.09 in Non-ex (*p* = 0.75)
		TCM Inject (admission) or HM (f/u) + WM	Wulin 2018 [34]	(During hospitalization) OR 1.27 in Non-ex (*p* = 0.09) (During follow-up) OR 1.09 in Non-ex (*p* = 0.51)
	Cardiac shock	TCM Inject (admission) or HM (f/u) + WM	Guohua 2018 [33]	(During hospitalization) OR 1.55 in Non-ex (*p* = 0.07)
		TCM Inject (admission) or HM (f/u) + WM	Wulin 2018 [34]	(During hospitalization) OR 1.97 in Non-ex (*p* < 0.001)
Adverse event/safety	Cardiac arrhythmia-related admission	Fuzi + HM	Tai 2022 [23]	140/921 (15.2%) vs. 204/921 (22.2%), HR = 1.03 (95% CI, 0.83–1.29)
	Dyspepsia	HM(SQLXF) + routine treatment	Sui 2018 [28]	5/75 (6.7%) vs. 4/60 (6.7%), *p* = 1.00
	Bloating	HM(SQLXF) + routine treatment	Sui 2018 [28]	4/75 (5.3%) vs. 2/60 (3.3%), *p* = 0.58
	Abdominal discomfort	HM(SQLXF) + routine treatment	Sui 2018 [28]	4/75. (5.3%) vs. 3/60 (6.7%), *p* = 0.93
	Abdominal pain	HM(SQLXF) + routine treatment	Sui 2018 [28]	2/75 (2.7%) vs. 2/60 (5.0%), *p* = 0.82
	Diarrhea	HM(SQLXF) + routine treatment	Sui 2018 [28]	1/75 (1.3%) vs. 0/60 (0.0%), *p* = 0.59
	In-hospital bleeding	TCM + WM	Yu 2019 [27]	167/7400 (2.3%) vs. 88/2509 (3.5%), OR = 1.11 (95%CI, 0.86–1.45)
Medical cost	Outpatient care medical cost (NT $)	TCM	Tsai 2017 [29]	1 year after HF: 53,602 (75,379) vs. 64,154 (140,817), *p* < 0.001 5 years after HF: 238,376 (427,151) vs. 222,161 (566,458), *p* = 0.71
	HF hospitalization cost (NT $)	TCM	Tsai 2017 [29]	1 year after HF 139,888 (265,767) vs. 243,614 (382,984), *p* = 0.02 5 years after HF 337,875 (603,776) vs. 315,813 (548,970), *p* = 0.70
	HF outpatient care visits (days)	TCM	Tsai 2017 [29]	1 year after HF: 49.43 (49.07) vs. 23.44 (19.99), *p* < 0.001 5 years after HF: 188.08 (202.39) vs. 70.12 (78.80), *p* < 0.001
	HF hospital length of stay (days)	TCM	Tsai 2017 [29]	1 year after HF: 39.63 (132.70) vs. 60.55 (209.06), *p* = 0.36 5 years after HF: 108.54 (368.48) vs. 134.62 (1097.03), *p* = 0.75
Cardiac function	NYHA classification improvement	SMI + WM + TCM	Guan 2022 [24]	No change) 161/451 (35.7%) vs. 337/596 (56.5%) 1 level improved) 273/451 (60.5%) vs. 243/596 (40.8%) 2 level improved) 17/451 (3.8%) vs. 16/596 (2.7%)
	LVEF (%) improvement	SMI + WM + TCM	Guan 2022 [24]	8.89 ± 10.72 vs. 7.91 ± 11.38 (*p*-value not presented)
		HM(SQLXF) + routine treatment	Sui 2018 [28]	6.7 (4.1–9.3) vs. 2.5 (1.1–3.8), *p* < 0.05
	LVDD	HM(SQLXF) + routine treatment	Sui 2018 [28]	−1.2 (−1.9–0.4) vs. −0.3 (−0.7–0.2), *p* = 0.23
	Cardiac output	HM(SQLXF) + routine treatment	Sui 2018 [28]	0.9 (0.4–1.3) vs. 0.3(0.1–0.5), *p* < 0.05
	Every cardiac output	HM(SQLXF) + routine treatment	Sui 2018 [28]	7.4 (4.9–9.0) vs. 2.3 (1.4–5.7), *p* < 0.05
	Cardiac index	HM(SQLXF) + routine treatment	Sui 2018 [28]	0.3 (0.1–0.6) vs. 0.2 (0.1–0.4), *p* = 0.31
	6 MWT	HM(SQLXF) + routine treatment	Sui 2018 [28]	172.6 (133.8–216.9) vs. 87.9 (51.4–118.7), *p* < 0.05
Laboratory	NT-proBNP	SMI + WM + TCM	Guan 2022 [24]	909 ± 3633 vs. 735 ± 3989 (*p*-value not presented)
Others	Predictive factors for TCM usage	TCM user	Tsai 2017 [30]	Women, young age, hypertension, coronary artery disease, chronic obstructive pulmonary disease

6 MWT, 6-min walking test; ACS, acute coronary syndrome; CCU, critical care unit; CI, confidence interval; CV, cardiovascular; f/u, follow-up; HF, heart failure; HM, herbal medicine; HR, (adjusted) hazard ratio; ICU, intensive care unit; LVEF, left ventricular ejection fraction; LVDD, left ventricular diastolic dysfunction; MI, myocardial infarction; NT $, New Taiwan Dollar; NT-proBNP, N-terminal pro-brain natriuretic peptide; NYHA, New York Heart Association; OR, Odds ratio; sHR, subdistribution hazard ratio; SMI, Shenmai injection; SQLXF, Shenqilixin formula; TCM, traditional Chinese medicine; WM, western medicine.

**Table 5 healthcare-12-00061-t005:** Components of prescribed botanical drugs (herbal medicines) in the included studies.

Scientific Name of Herbal	Pinyin	Tai 2022 [23] (Taiwan)	Guan 2022 [24] (China)	Sui 2018 [28] (China)	Guohua 2018 [33] (China)	Wulin 2018 [34] (China)	Yu 2019 [27] (China)	Huang 2019 [31] (China)	Liu 2022 [32] (Taiwan)	Tsai 2017 [29] (Taiwan)	Tsai 2017 [30] (Taiwan)
*Panax ginseng* C. A. Meyer	Renshen	O	O	O	O	O	O			O	O
*Astragalus membranaceus* Bunge	Huangqi			O	O	O	O		O	O	O
*Salvia miltiorrhiza* Bunge	Danshen			O	O	O	O		O	O	O
*Glycyrrhiza uralensis* Fisch.	Gancao	O		O	O	O				O	O
*Paeonia lactiflora* Pall.	Chishao	O			O	O				O	O
*Ophiopogon japonicus* Ker-Gawler	Maidong		O		O	O				A *	O
*Panax notoginseng* (Burk.) F. H. Chen *Radix pseudoginseng*	Sanqi TianSanqi				O	O	O			O	O
*Carthamus tinctorius* L.	Honghua				O	O	O			O	O
*Ligusticum chuanxiong* Hort.	Chuanxiong				O	O			O	O	O
*Aconitum carmichaelii* Debx.	Fuzi	O				O				O	O
*Poria cocos* (Schw.) Wolf	Fuling	O		O						O	O
*Paeonia lactiflora* Pall.	Baishao	O				O				O	O
*Zingiber officinale* Rosc.	Ganjiang	O				O				O	O
*Rehmannia glutinosa* (Gaertner) Liboschitz ex Steudel	Dihuang				O	O				O	O
*Cinnamomum cassia* Presl	Guizhi			O		O				O	O
*Schisandra chinensis* (Turcz.) Baill.	Wuweizi				O	O				O	O
*Angelica sinensis* (Oliv.) Diels	Danggui				O	O				O	O
*Platycodon grandiflorum* (Jacq.) A.DC.	Jiegeng				O	O				O	O
*Prunus persica* (L.) Batsch	Taoren				O	O				O	O
*Cyathula officinalis* Kuan	Chuanniuxi				O	O				O	O
*Bupleurum chinense* DC.	Chaihu				O	O				O	O
*Atractylodes macrocephala* Koidz.	Baizhu	O								O	O
*Scutellaria baicalensis* Gerogi	Huangqin				O			O	O		
*Equus asinus* Linne	Ejiao					O				O	O
*Ziziphus jujuba* Mill.	Dazao					O				O	O
*Zingiber officinale* Rosc.	Shengjiang					O				O	O
*Ziziphus jujuba* Mill. var. spinosa (Bge.) Hu ex H.F.Chou	Suanzaoren					O				O	O
*Rehmannia glutinosa* Liboschitz ex Steudel	Shudihuang					O				O	O
*Alisma orientale* (Sam.) Juzep.	Zexie					O				O	O
*Cinnamomum cassia* Presl	Rougui					O				O	O
*Panax ginseng* C. A. Meyer	Hongshen					O	O				
*Ginkgo biloba* Linné	Baiguo					O	O				
*Agrimonia pilosa* Ledebour	Xianhecao			O			O				
*Curcuma longa* Linné	Jianghuang					O		O			
*Cannabis sativa* Linné	Huomaren									O	O
*Anemarrhena asphodeloides* Bunge	Zhimu									O	O
*Ligusticum sinense* Oliv.	Gaoben									O	O
*Asparagus cochinchinensis* (Lour.) Merr.	Taindong									O	O
*Scrophularia buergeriana* Miquel	Xuanshen									O	O
*Polygala tenuifolia* Willd.	Yuanzhi									O	O
*Platycladus orientalis* (L.) Franco	Baiziren									O	O
- (Botanical name: Cinnabaris)	Zhusha									O	O
*Citrus aurantium* Linné	Zhiqiao									O	O
*Cornus officinalis* Siebold et Zuccarini	Shanzhuyu									O	O
*Dioscorea opposita* Thunb.	Shanyao									O	O
*Paeonia suffruticosa* Andr.	Mudanpi									O	O
*Citrus reticulata* Blanco	Qingpi									O	O
*Cimicifuga heracleifolia* Kom.	Shengma									O	O
*Glycyrrhiza uralensis* Fisch.	Zhigancao					O					O
*Codonopsis pilosula* (Franch.) Nannf.	Dangshen					O					O
*Trichosanthes kirilowii* Maxim.	Gualou				O						O
*Dryobalanops aromatica* Gaertner	Bingpian				O	O					
*Dalbergia odorifera* T. Chen	Jiangxiang				O	O					
*Poncirus trifoliata* Rafinesque	Zhishi				O	O					
*Allium macrostemon* Bge.	Xiebai				O	O					
*Pinellia ternata* Breitenbach	Banxia				O	O					
*Ephedra sinica* Stapf	Mahuang	O									
*Asarum sieboldii* Miq. var. seoulense Nakai	Xixin	O									
*Epimedium brevicornum* Maxim	Yinyanghuo			O							
*Lepidium apetalum* Willdenow	Tinglizi			O							
*Atractylodes chinensis* (DC.) Koidz.	Changzhu			O							
*Leonurus japonicus* Houttuyn	Yimucao			O							
*Hedyotis diffusa* Willdenow	Baihuasheshecao							O			
*Taraxacum mongolicum* Handel-Mazzetti	Pugongying							O			
*Hordeum vulgare* L.	Maiya				O						
*Moschus berezovskii* Flerove	Shexiang					O					
*Pheretima aspergillum* (E.Perrier)	Dilong					O					
*Eupolyphaga sinensis* Walker	Tubiechong					O					
*Equus asinus* Linne	Ejiao					O					
*Scolopendra subspinipes mutilans* L. Koch	Wugong					O					
*Buthus martensii* Karsch	Quanxie					O					
*Pinellia ternata* Breitenbach	Banxia										O
*Poria cocos* Wolf	Fushen										O
*Plantago asiatica* Linné	Chenqianzi									O	
*Uncaria sinensis* Havil.	Gouteng								O		
*Rheum palmatum* L.	Dahuang								O		
*Pueraria lobata* Ohwi	Gegen								O		
*Eucommia ulmoides* Oliv.	Duzhong								O		
*Prunella vulgaris* L.	Xiakucao								O		
*Crataegus pinnatifida* Bge.	Shanzha								O		

The prescribed herb in each study is denoted by an ‘O’. A * Liriope Tuber.

**Table 6 healthcare-12-00061-t006:** Botanical drugs (herbal medicines) used in the included studies.

Study ID	Intervention (Formulation)
Tai 2022 [23]	Any type of herbal prescription, including Fuzi
Guan 2022 [24]	Shenmai injection
Komagamine 2021 [25]	Kampo medications that can cause or exacerbate heart failure
Yu 2019 [27]	Danshen dripping pills, Danshen polyphenolate injection, etc. Xueshuantong injection, Sanqi Tonghu capsule, etc. Qiliqiangxin capsule, Qishenyiqi Dripping pills, etc. Ginkgo leaf capsule; Ginkgo leaf extract and dipyridamole injection, etc. Safflower yellow injection; Safflower extraction, etc.
Huang 2019 [31]	Not applicable
Sui 2018 [28]	Shen Qi Li Xin Formula
Guohua 2018 [33]	Shengmai, Dahong, Ligustrazine injection (main treatment) + herbal medicine
Wulin 2018 [34]	Shengmai, Danhong, Ligustra injection + herbal medicine
Tsai 2017 [29]	Zhi-Gan-Cao-Tang, Sheng-Mai-San, Zhen-Wu-Tang, Suan-Zao-Ren-Tang, Tian-Wang-Bu-Xin-Dan, Xue-Fu-Zhu-Yu-Tang, Ji-Sheng-Shen-Qi-Wan, Bu-Zhong-Yi-Qi-Tang, Si-Ni-Tang, Liu-Wei-Di-Huang-Wan
Tsai 2017 [30]	Zhi-Gan-Cao-Tang, Sheng-Mai-San, Zhen-Wu-Tang, Suan-Zao-Ren-Tang, Bu-Zhong-Yi-Qi-Tang, Si-Ni-Tang, Tian-Wang-Bu-Xin Dan, Xue-Fu-Zhu-Yu-Tang, Liu-Wei-Di-Huang-Wan, Yang-Xin-Tang
Liu 2022 [32]	Tian-Ma-Gou-Teng-Yin, Xue-Fu-Zhu-yu-Tang, Gou-Teng-San, Jia-Wei-Xiao-Yao-San, Zhi-Gan-Cao-Tang, Ji-Sheng-Shen-Qi-Wan, Liu-Wei-Di-Huang-Wan, Zhi-Bai-Di-Huang-Wan, Bu-Yang-Huan-Wu-Tang, Qi-Ju-Di-Huang-Wan

## Data Availability

Data that support the findings of this study are available from the corresponding authors upon reasonable request.

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
