# Peer review of "Current Research Status and Implication for Further Study of Real-World Data on East Asian Traditional Medicine for Heart Failure: A Scoping Review"

_healthcare, 2023, doi:10.3390/healthcare12010061_

Round 1

Reviewer 1 Report

Comments and Suggestions for Authors

The manuscript "Current research status and future perspective ...." by Park et al. describes findings on the clinical use of East Asia traditional medicine for heart failure. Comments are as follow:

1) Title: The study is good, but it should include data on future perspectives to match the title's claim.

2) Methodology: Consider revising the research questions in Step 1 into a paragraph.

3) Lines 156-162: The table titles should not be enclosed in inverted commas. Tables should be placed under the "Results" section, and the list of abbreviations under tables should be arranged alphabetically.

4) Terminology: Replace "herbal medicines" with "botanical drugs." Provide scientific names (species and authority name), routes of administration, and composition for mixture preparation. This information should be in the main text, not in supplementary table(s).

5) Table 3: The findings were with intervention? Clarify the meaning of "O."

6) Discussion: Avoid subtopics in the discussion section.

7) References: Increase the proportion of references from the last 5 years (2018-2023) to 70-80%.

Comments on the Quality of English Language

Minor correction needed.

Reviewer 2 Report

Comments and Suggestions for Authors

I’ve read with attention the paper of Park et al. that is potentially of interest. The background and aim of the study have been clearly defined. The methodology applied is overall correct, the results are reliable and adequately discussed. I’ve only some minor comments:

- The abstract should contain some quantitative data or at least an example of effective treatment detected by the literature review

- The discussion should point out that traditional medicines should be not necessarily cheaper nor safer than the conventional one and that patients assuming traditional remedies often also take standard drugs (with possible pharmacological interaction as well as possible positive synergies).

- Among limitations, it should be stressed that the reported analysis has been restricted to East-Asian studies, and not directly applicable to different settings.

Comments on the Quality of English Language

The paper should be attentively revised as regards English language because some sentences sound not idiomatic or "unusual".

Reviewer 3 Report

Comments and Suggestions for Authors

The authors had done a thorough scoping review. However, the review was found focused to herbal medicine which is found inconsistent with the title as the title included with EATCM , in which TCM also included with other treatment methods including acupuncture, cupping , guasha, moxibustion, tuina or massage, TCM chiropractric. 

Suggest to revise the title to reflect the content of review 

Introduction

The introduction had included with detailed on the lead factor to heart failure, complications of HF and the treatment for HF. 

However, the introduction can be accomplished with more detailed on the pathogenesis or aetiology or mechanism on the lead factors to heart failure. 

Methods 
Detailed procedure and study selection was included. 

-review can include the keywords used for journal search for the records (n=258)
-- figure 1 need to be verified as 258 from the records but after removed 51 records, it should left 207 which is shown inconsistency 

- figure 1 --DB should included the footnote or full term 

- figure 1 --incomplete as the final study (n=12) was not included 

-Table 1 - title can be revised as the title was sound hanging 

(same to Table 2-4 )

Result 

Detailed result was included but there is not much in relation to the pharmacology or mechanism of action of the herbal medicine that being used

Discussion

Suggest to include the mechanism of action of the herbal medicine of the earlier study and it may also review on the cost of treatment .

There are redundancy to 314-323 and 342 -361, suggest to be combined or revised. 
